# Iodine increases pulmonary type I interferon responses and decreases covid-19 disease severity: Results from an open label randomized clinical trial

René Traksel[1]*, Jasper Broen [1], Arjen van Henten[2], Marc Königs[3], Arjun Raj[4], Laura van Eyndhoven [4,5], Richard Verheesen[1]

**1** Regional Rheumatology Center, Maxima Medical Center Eindhoven, The Netherlands, **2** Pulmonary medicine, Maxima Medical Center Veldhoven, The Netherlands, **3** Intensive care medicine, Maxima Medical Center Veldhoven, The Netherlands, **4** Department of Bioengineering, School of Engineering and Applied Sciences, University of Pennsylvania, Philadelphia, Pennsylvania, United States of America, **5** Department of Biomedical Engineering, Eindhoven University of Technology, Eindhoven, The Netherlands

☯ These authors contributed equally to this work.
* r.traksel@mmc.nl

## Abstract

### Objective

To investigate whether oral treatment with 12.5 mg iodine additional to standard of care is effective in reducing mortality and clinical deterioration of patients hospitalized with COVID-19.

### Methods

We performed a single center, randomized clinical trial (EudraCT 2020-001852-16) in which patients with severe covid-19 in need of hospitalization were randomized in two groups. The first group received 12.5 mg oral iodine for 8 days, the second group did not receive iodine next to the standard of care. Primary endpoints were deterioration of disease defined as transfer from the ward to the intensive care unit (ICU) or death. Next to these parameters we collected parameters in line with the recommendations made by the WHO in the early days of the pandemic. On these additional datasets we performed an exploratory analysis and investigated possible confounders and trends. The inclusion phase of the study was between October 2020 and April 2022. Finally, *in vitro* validations were performed.

### Results

Outcomes from 141 participants were analyzed, revealing no significant differences in mortality or transfers to intensive care between the iodine-treated group (67 patients) and the control group (74 patients). In an exploratory analysis we found

**Data availability statement:** All relevant data are within the manuscript and its Supporting Information files.

**Funding:** This study has been performed with financial aid of the Maxima Medical Center Innovation Fund. The funders had no role in study design, data collection and analysis, decision to publish, or preparation of the manuscript.

**Competing interests:** The authors have declared that no competing interests exist.

that patients randomized to receive oral iodine had a significantly shorter stay at the ICU ($p = 0.016$). *In vitro* validations proved increased virus-induced type I interferon responses upon iodine administration in pulmonary cells.

## Conclusion

These findings suggest that while iodine does not reduce mortality or ICU admissions, it may enhance antiviral immunity through increased type I interferon responses, contributing to shorter ICU stays in COVID-19 patients. The role of iodine in enhancing IFN-I mediated antiviral immunity warrants future research.
Registration of trial: EudraCT Number: 2020-001852-16. https://www.clinicaltrialsregister.eu/ctr-search/search?query=2020-001852-16
Sponsor name: Maxima Medical Center. Date of Registration: April 1st 2020.

## Introduction

At the end of 2019 a new coronavirus was discovered, nowadays well-known as SARS-CoV-2. The virus rapidly spread across the world leading to a pandemic with many individuals hospitalized and deceased by the subsequent disease COVID-19. The large number of deaths and hospitalization put a heavy burden on healthcare and society, sparking many efforts to find a treatment that would lead to lower mortality and hospitalization. There is a large, still unexplained, difference between individuals in the extent to which one is affected by the virus. Some people do not notice the virus, while others need intensive care treatment. This observation sparked a light on the role of diet and lifestyle on COVID-19 disease susceptibility and disease course. One of these avenues of diet and lifestyle research focused on micronutrients and vitamins, for instance witnessed by the large number of studies describing the effect of vitamin D intake in COVID-19 [1].

In line with the raised interest on the role of diet and micronutrients in COVID-19, we postulated the hypothesis that oral iodine treatment might reduce disease burden in COVID-19 disease [2]. Oral iodine administration has a long tradition in the treatment of influenza and other viral infections of the respiratory tract [2–5]. This alleged beneficial attribute of iodine, however, received little contemporary scientific attention and clinical trials investigating the effect of oral iodine on infections of the respiratory tract are lacking. Although clinical trials are absent, there is evidence from *in vivo* animal and *in vitro* human research that iodine indeed exerts an anti-viral effect on pulmonary infections [6–8]. For instance, it has been shown that administering iodine to lambs resulted in higher levels of iodide in the airway secretions leading to lower virus expression and pulmonary lesions when these lambs were subsequently inoculated with the respiratory syncytial virus [8]. *In vitro* research with human peripheral blood lymphocytes, shows that iodine might enhance the ability of lymphocytes to produce immunoglobulin G (Ig), of which production is pivotal to mount an adequate anti-viral response [9].

Another observation that prompted us to perform a randomized clinical trial with adding oral iodine to the standard of care for hospitalized COVID-19 patients is the fact that in Japan, which has one of the oldest and most dense populations, the disease burden of COVID-19 was one of the lowest. The cause of this somewhat counterintuitive finding is still enigmatic today [10]. Since the Japanese, however, have the highest dietary iodine intake in the world, we found this observation to be another reason to further substantiate the effect of oral iodine administration on COVID-19 disease burden.

Taking together these observations from historic reports, immunological research, low COVID-19 burden in high iodine intake areas and pressing need for a low-cost effective treatment of COVID-19, we set out on an endeavor to investigate a possible beneficial effect of oral iodine administration as addition to standard of care for COVID-19 patients in a hospitalized setting. In addition, we performed *in vitro* validations in lung epithelium cells and leukocytes.

## Materials and methods

### Patients and recruitment

Patients with age above 18 admitted to Maxima Medical Center (Community Hospital, Veldhoven, The Netherlands) between October 1, 2020, and April 1, 2022, with PCR confirmed COVID-19 were invited to participate. Patients were approached within the first 72 hours of hospital admission and needed to be able to give informed consent themselves. Patients with thyroid medication or amiodarone use were excluded from the study. All patients received standard of care treatment for COVID-19 disease congruent with the stage of the pandemic. This included prednisolone, hydroxychloroquine, tocilizumab and sarilumab at different stages. Before the study was initiated it was approved by the local ethical committee (METC Brabant, Netherlands no NL73784.028.20) and the study complies fully with the Helsinki declaration (EudraCT 2020-001852-16).

### Study medication

Participants were automatically randomized into intervention (standard care + oral iodine) and control (standard care) groups. The intervention group received potassium-iodine tablets containing 12.5 mg iodine, once daily for 8 days. The tablets were administered orally or, in case of worsening of symptoms and transport to the intensive care unit, administered through the nasal-gastro-intestinal tube. The tablets were produced by Gerot Lanach, lotnumber 6C183A. The control group did not receive a *placebo*.

### Data collection and analysis

We performed a single center, randomized, open label, parallel group clinical trial. Randomization was automatically performed with the program Research Manager. As primary outcome measures, we defined death and deterioration of symptoms leading to transfer from general ward to the intensive care unit. In addition, we collected extensive data as was advised at the early stages of the pandemic by the World Health Organization. More specifically the data was collected with the standardized WHO core Case Report Form (CRF), consisting of three modules for admission, ICU care and discharge [11]. Data collection was performed with Research Manager Software. Subsequent analyses were performed with SPSS v. 16.0 and figures were created with Sankymatic and SPSS. For continuous variables we used t-tests and linear regression and for non-continuous variables we used chi-square tests or Fisher exact tests where appropriate. We did not encounter missing data. Since the analyses performed on the non-primary outcomes were exploratory and merely intended to provide possible avenues for further research, we did not perform correction for multiple testing on these outcomes.

### Cell culture and in vitro activation

Calu-3 (HTB-55) cells were cultured at 37°C and 5% $CO2$ in growth medium consisting of MEM Earles with Glutamax (Thermo Fisher Scientific, 11095114) + 10% FBS (Thermo Fisher Scientific, 16000044) 1% penicillin/streptomycin

(Invitrogen, 15140122). Cells were split 1:5 upon reaching 80% confluency using 0.05% trypsin-EDTA (Invitrogen, 25300054). For in vitro validation experiments, cells were seeded in 24-well glass-bottom plates 24 hours before transfection. Cells were transfected with 1 µg/mL Poly(I:C) using Lipofectamine2000 transfection reagent (Invitrogen, 12566014) according to manufacturer's instructions. Rhodamine-labeled Poly(I:C) (InvivoGen, tlrl-piwr) was used for transfection control validations.

## smRNA FISH

single-molecule RNA FISH was performed as previously [12]. For all genes of interest, complementary oligonucleotide probe sets were designed using a custom probe design software which excludes probes that are complementary to untargeted genes (MATLAB). Probes with a primary amine group on the 3' end were ordered from Biosearch Technologies. Probes were pooled and coupled to Cy3 (GE Healthcare), Alexa Fluor 594 (Life Technologies) or Atto647N (ATTO-TEC) N-hydroxysuccinimide ester dyes. Cells were fixed in 4% formaldehyde for 10 minutes at room temperature and permeabilized in 70% ethanol for 1 hour at room temperature. Next, samples were washed with 2x SSC and 10% formamide before hybridization with custom RNA FISH probes in 2x SSC, 10% formamide, and 10% dextran sulfate overnight at 37°C. Next, samples were washed twice for 30 minutes at 37°C and stained with DAPI. Samples were imaged in 2x SSC on Nikon Ti-E at 60X. Nikon Elements imaging software was used to capture multi-tile images. Nikon Perfect Focus System ensured all tiles were equally in focus and a 7-step z-stack ensured all RNA spots were captured. The sample size was calculated based on an expected deterioration rate of 25% in the control group and 10% in the intervention group, with a power of 80% and a two-sided significance level of 0.05. This resulted in a required sample size of 100 participants per group.

## Image analysis

Image analysis was performed using NimbusImage, an open-source image analysis platform developed by the lab of Arjun Raj at the University of Pennsylvania (https://github.com/Kitware/UPennContrast). DAPI-stained nuclei were segmented using Cellpose, a generalist algorithm for cellular segmentation [13]. RNA spots were identified using Piscis, a deep-learning algorithm for spot detection (https://github.com/zjniu/Piscis). Piscis was trained on representative images for accurate spot detection [14]. RNA spots were assigned to nuclei using the Connect to Nearest tool, which connects objects based on distance. Image annotations were exported as CSV files for data visualization in GraphPad Prism.

## Results

### Patient characteristics

Overall, 150 patients with proven COVID-19 disease that needed admission to the hospital were included in the trial and computer block (blocks of 10) randomized in two groups; the first group receiving only standard of care and the second group receiving standard of care + 12,5 mg iodine once daily during five days. The personnel had no access to randomized allocation sequence. Within the standard care + iodine group, nine patients were eventually excluded because they did not finish the full eight days of treatment. Important to mention here is that this was not due to rapid deterioration of disease but due to logistic issues such as outplacement to another hospital or overloaded nursing staff during this hectic phase of the pandemic. **Table 1**. shows the characteristics of the 74 patients included in the standard of care group and those of the 67 patients in the standard of care + iodine group. There is a significant difference in the proportion of patients using vitamin K antagonists in favor of the standard of care + iodine group ($p = 0.02$). However, we did not see any differences in accepted risk factors for severe COVID-19 such as body mass index, sex, cardiovascular or pulmonary disease. Moreover, no difference in vaccination status for Sars-Cov-2 was observed between the two groups. No serious events related to the iodine treatment were observed, on the other hand several severe adverse events were recorded caused by the COVID-19 disease itself. This did not differ between the two study populations.

**Table 1. Characteristics of included patients at inclusion stratified to randomization.**

|  | No iodine treatment | Iodine treatment |
|---|---|---|
| **n** | 74 | 67 |
| **Female** | 0.32 | 0.33 |
| **Age** | 69.7 (SD 12.9) | 71.2 (SD 11.9) |
| **Body temperature at admission** | 38 (SD 1) | 37.9 (SD 1.1) |
| **Systolic Bloodpressure** | 137 (SD 23) | 142 (SD 24) |
| **Diastolic Bloodpressure** | 74 (SD 12) | 77 (SD 11) |
| **Heart rate** | 90 (SD 18) | 89 (SD 17) |
| **Respiratory rate** | 28 (SD 9) | 27 (SD 12) |
| **Oxygen saturation** | 89 (SD 7) | 89 (SD 9) |
| **Hypertension** | 0.34 | 0.48 |
| **BMI** | 27 (SD 6) | 28 (SD 5) |
| **Chronic heart disease** | 0.3 | 0.3 |
| **Chronic pulmonary disease** | 0.27 | 0.38 |
| **Chronic liver disease** | 0.01 | 0.01 |
| **Chronic neurological disorder** | 0.15 | 0.26 |
| **Chronic kidney disease** | 0.74 | 0.70 |
| **eGFR > 90** | 0.26 | 0.33 |
| **eGFR 89−60** | 0.45 | 0.37 |
| **eGFR 59−45** | 0.17 | 0.16 |
| **eGFR 44−30** | 0.11 | 0.1 |
| **eGFR 29−15** | 0.01 | 0.04 |
| **eGFR < 15** | 0.00 | 0.00 |
| **Diabetes** | 0.24 | 0.23 |
| **Smoking** | 0.13 | 0.07 |
| **Malignant neoplasm** | 0.20 | 0.09 |
| **ACEi** | 0.19 | 0.26 |
| **ARBs** | 0.17 | 0.12 |
| **NSAIDs** | 0.04 | 0.03 |
| **Glucocorticoids** | 0.06 | 0.11 |
| *Immunomodulatory drugs other than glucocorticoids* | 0.08 | 0.09 |
| **Vitamin D supplements** | 0.45 | 0.43 |
| **Other vitamin supplements** | 0.30 | 0.21 |
| **Vitamin K antagonists** | 0.01 | 0.10* |
| **NOAC** | 0.23 | 0.13 |
| **acetylsalicytic acid** | 0.19 | 0.19 |
| **Influenza vaccination previous 12 months** | 0.50 | 0.51 |
| **SARS-cov-2 vaccine** | 0.49 | 0.54 |

n = number, BMI = Body Mass Index, eGFR = estimated glomerular filtration rate, ACEi- angiotensin converting enzyme inhibitor, ARB = angiotensin receptor blocker, NSAIDs = nonsteroidal anti-inflammatory drugs, NOAC = new oral anticoagulant agents, SARS-CoV-2 = Severe acute respiratory syndrome coronavirus 2, * = significant at the p < 0.05 level.

## Primary outcomes

Our primary outcomes were (I) dead and (II) worsening of disease leading to intensive care unit admission. We did not observe any significant differences in these outcomes between the two groups. This remained unchanged after correcting for vitamin K antagonist use. The numbers of patients reaching these endpoints from inclusion are depicted in **Fig 1** and **Table 2**.

## Exploratory analyses

As discussed in the methods section we collected all data in accordance with the CRF developed under the guidance of the WHO. In this CRF abundant data is collected about treatments, physical and biochemical markers, oxygen therapy, invasive oxygen therapy and so on. The study was not designed to analyze these additional parameters. We felt, however, in the light of the pressing need for treatments during the pandemic, that performing an exploratory analysis on this data was a prudent addition to the study possibly leading to new insights.

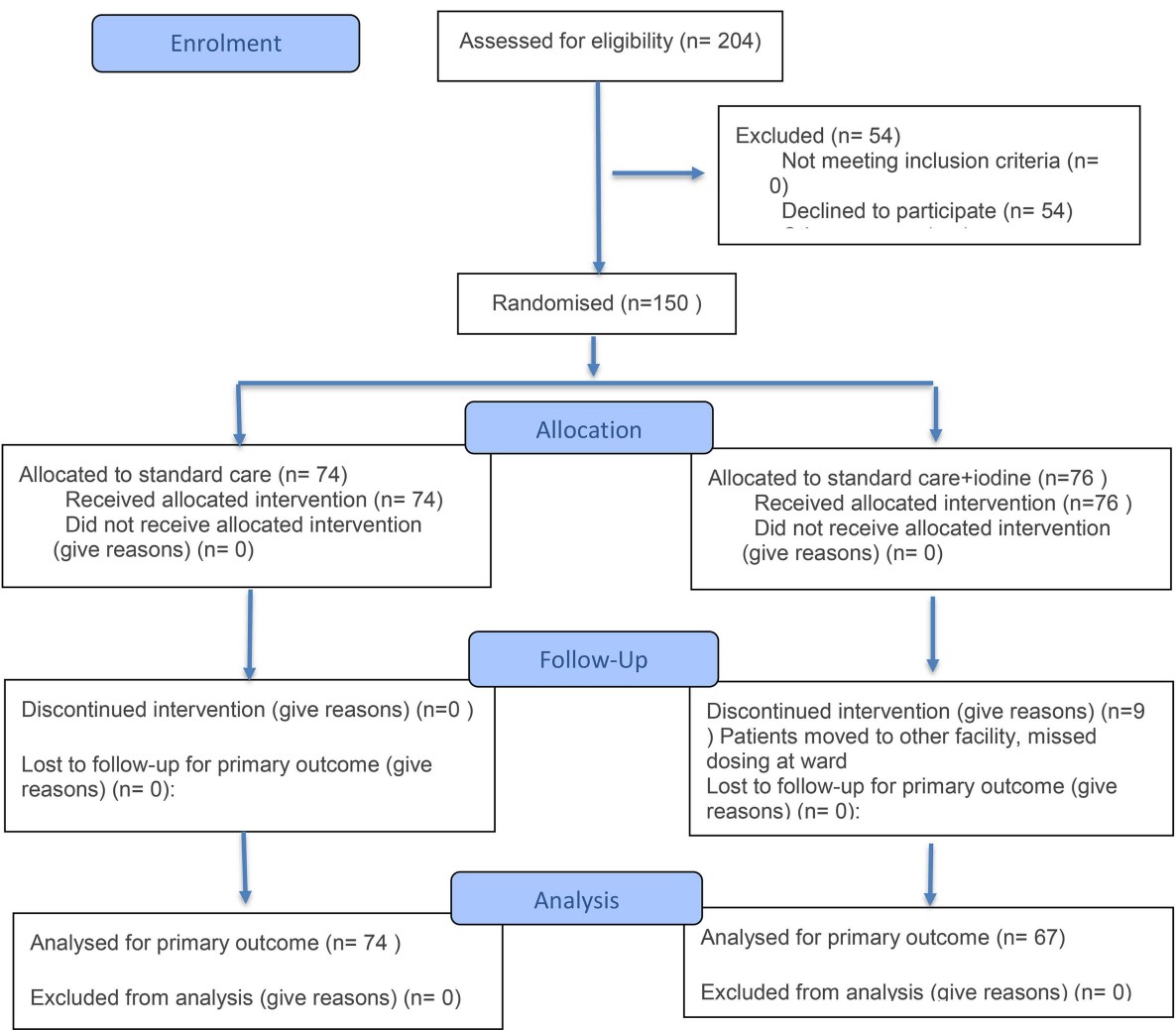

**Fig 1. Patient inclusion flowchart with primary endpoint outcomes.**

**Table 2. Outcomes of randomized clinical trial.**

| Outcome | No iodine treatment (n = 74) | Iodine treatment (n = 67) |
|---|---|---|
| Death | 0.18 | 0.19 |
| Days from first dose to death | | 11. 3 (SD 17.6) |
| Days from first dose or study inclusion to discharge alive | | 11.09 (SD 25) |
| Days from admission to death | 5.85 (SD 3.7) | 11.85 (SD 18) |
| Days from admission to discharge alive | 13.5 (SD 17) | 12 (SD 25) |
| Adverse events | 0.66 | 0.70 |
| Serious adverse events | 0.35 | 0.31 |
| Transfer to intensive care unit | 0.15 | 0.10 |
| Total days admission intensive care unit | 26.5 (SD 19) | 8.33 (SD 5.5)* |
| Oxygen therapy % | 0.97 | 0.98 |
| Days of oxygen therapy | 11 (SD 16) | 9 (SD 10) |
| Invasive ventilation % | 0.14 | 0.09 |
| Days of invasive ventilation | 21 (SD 15) | 17 (SD 23) |
| Extra Corporal Membrane Oxygenation | 0 | 0 |
| Renal replacement therapy | 0.00 | 0.01 |
| Inotropic therapy | 0.04 | 0.00 |
| CT scan with iodine contrast | 0.49 | 0.55 |

* $p < 0.05$, statistical tests used include t-tests, chi-square tests, and Fisher's exact tests.

Numbers are rounded to two decimal places..

In these exploratory analyses comparing additional outcome parameters between the standard of care and the standard of care + iodine group we observed a significant difference in time of ICU treatment ($p = 0.016$). In which the patients that received iodine (8.33 days S.D. 5.5, $n = 7$) had a significantly shorter stay on the ICU compared to the standard of care group (26.5 days S.D. 19, $n = 11$) (**Fig 2**). No confounders of this effect were revealed by additional analyses. Furthermore, no additional significant differences between the two groups were present (**Table 2**).

## In vitro validation

Intrigued by this clinical observation, we aimed to validate this finding in an *in vitro* context. To mimic viral infection, we activated Calu-3 cells, an adenocarcinoma lung cell line, with polyinosinic:polycytidylic acid (poly(I:C)), a synthetic viral ligand encapsulated in lipid-based nanoparticles for efficient transfection. Poly(I:C) functions as a synthetic, non-replicating virus, making it an ideal tool for quantifying antiviral immune responses *in vitro* [15].

Antiviral immunity against SARS-CoV-2 is predominantly mediated by the production of pro-inflammatory cytokines and type I interferons (IFN-I) [16]. Especially the timing of IFN-I production has been linked to the disease severity, with an early IFN-I response correlating with rapid viral clearance and milder disease outcomes. When timely, IFN-I can be secreted by early infected cells, subsequently activating antiviral immunity in neighboring, yet uninfected cells via the induction of interferon-stimulated genes (ISGs) [17]. Conversely, delayed IFN-I production is associated with viral persistence, chronic inflammation, and cytokine storms, contributing to severe disease [18]. With our clinical observation in mind, we hypothesized that iodine could enhance an early and robust IFN-I response.

To capture a fast and potent IFN-I response *in vitro*, we leveraged single molecule RNA fluorescence in situ hybridization (smRNA FISH) to quantify early single-cell antiviral behaviors, primarily focusing on IFNβ expression. To validate IFNβ translation and secretion, we additionally quantified the expression of ISGs, interferon-induced protein with tetratricopeptide repeats 1 (IFIT1) and Myxovirus resistance 1 (MX1), which are upregulated upon IFN-I receptor activation and serve

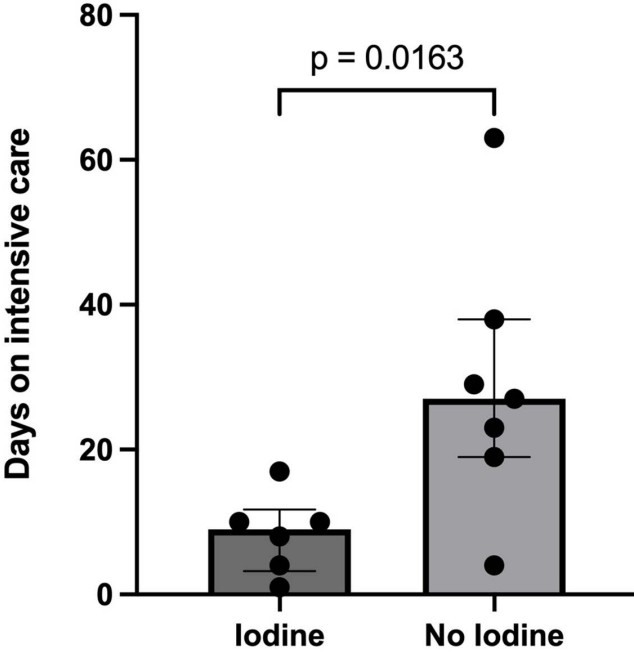

**Fig 2. Boxplot and whiskers of duration of stay at the intensive care unit, which shows a significant difference between standard care (n = 11) and standard care + iodide (n = 7).**

as markers of IFN-I signaling [16]. Additionally, IFIT1 and MX1 expression generally correlate with an enhanced antiviral state, contributing to protection against viral replication and disease progression [19,20].

Calu-3 cells activated with Poly(I:C) for 6 hours show highly heterogeneous antiviral responses, with only a subset of cells expressing IFNβ (Fig 3A). This heterogeneity is consistent with recent single-cell studies highlighting the vast degree of variability in single-cell IFNβ expression across cell types and experimental conditions [21]. The latter we validated with rhodamine-labeled Poly(I:C) transfection optimizations. To investigate whether iodine modulates antiviral immunity, we pretreated cells with Lugol's solution, a ready-to-use aqueous solution containing 0.7% potassium iodide and 0.35% iodine dissolved in water (Sigma-Aldrich, 32922). We treated the cells at a concentration range of 0.01%−1% (v/v) Lugol's solution in culture medium, which corresponds to a iodine concentration of 0.35-35 µg/mL. Notably, treatment with 1% Lugol solution markedly enhanced IFNβ expression following activation. This enhancement was seen not only in the number of cells producing IFNβ but also in the amount of IFNβ RNA produced per cell (Fig 3B). The increased IFNβ expression was mirrored by elevated IFIT1 and MX1 levels, indicating that Lugol treatment promoted not only increased IFNβ transcription but also its translation, secretion, and downstream signaling (Fig 3C). Importantly, Lugol treatment did not affect transfection efficiencies (S1 and S2 Fig) and did not induce IFNβ, IFIT1 or MX1 expression in unactivated cells (S3 Fig).

In conclusion, our *in vitro* validation supports a role for iodine in modulating IFN-I responses. Specifically, iodine appears to enhance early IFN-I production upon viral recognition. We therefore hypothesize that supplemental iodine in patients with severe COVID-19 symptoms could lead to a faster and more effective IFN-I response in yet uninfected lung epithelial cells, potentially slowing disease progression and reducing ICU stays.

## Discussion

In this study we analyzed the effect of adding oral iodine administration to standard care for severe COVID-19 disease, requiring hospitalization. Our primary hypothesis was that oral iodine reduces mortality and reduces transfers to the

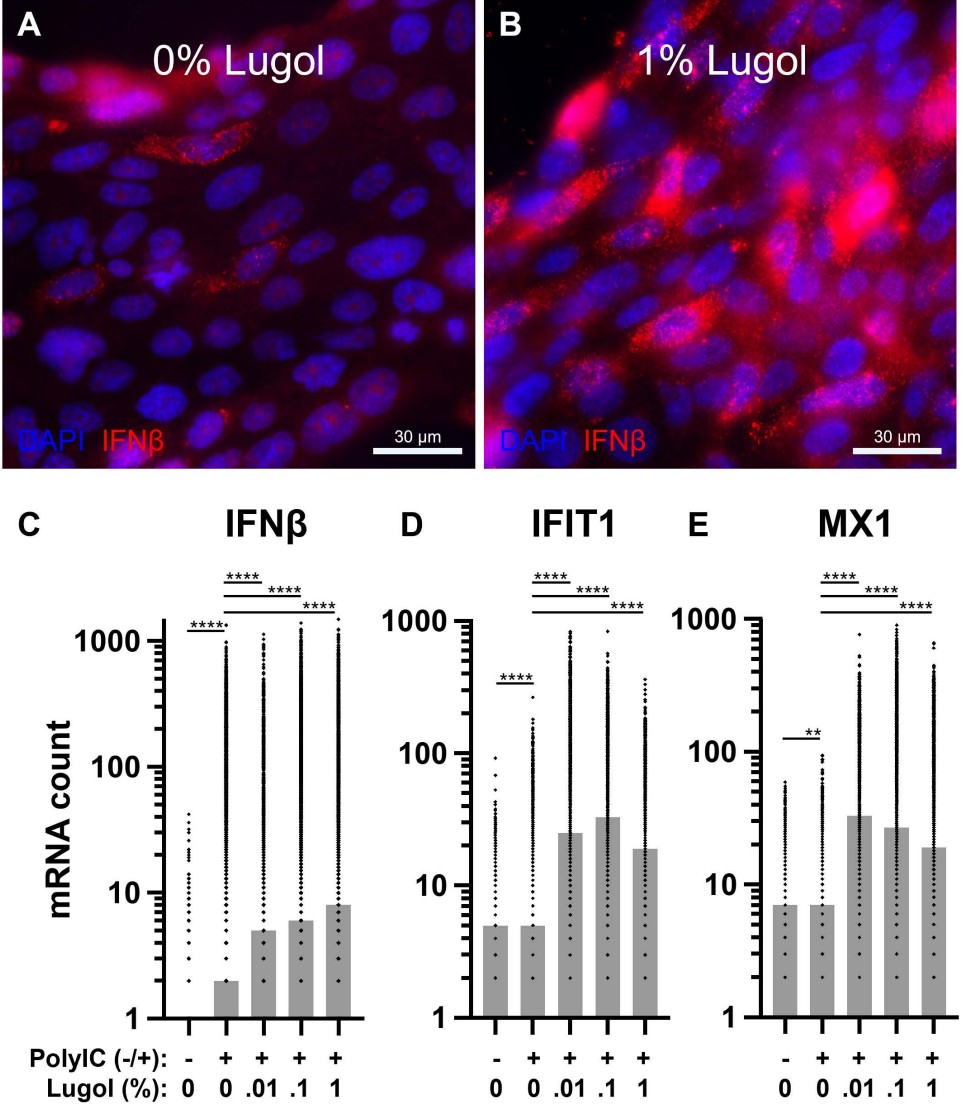

**Fig 3. Lugol increases IFNβ and subsequent ISG expression upon PolyIC activation in Calu-3 lung epithelium. A** Representative microscopy image of Calu-3 lung epithelium activated with PolyIC for 6 hours. IFNβ expression was monitored using smRNA FISH. **B** Representative microscopy image as in **A**. Cells were treated with 1% Lugol solution for 7 hours in total, starting 1 hour prior activation with PolyIC for 6 hours. **C** Quantification of smRNA FISH data for IFNβ. **D** Quantification of smRNA FISH data for IFIT1. **E** Quantification of smRNA FISH data for MX1. Bars represent median values per condition, dots represent single cells. Mann-Whitney test, n = 4, p** < 0.01, p**** < 0.0001.

intensive care unit. Our study does not corroborate this hypothesis and primary endpoints were not met. This finding indicates that adding this treatment in the armamentarium for severe COVID-19 is less likely to be relevant in reducing death and clinical deterioration in a hospital setting. Notably, due to the pandemic coming to an end we were not able to reach the target size, which might have played a role in failure to find a significant effect on mortality or ICU transfer. However, a strength of this study is the design as a randomized clinical trial, which is the first of its kind. Other strengths are the relatively large number of participants and the collection of data that has been done in a standardized way as was recommended by the WHO during the early stages of the pandemic. This abundancy of clinical data made it possible to

rule out most confounding factors arising from population heterogeneity, as well as differences in treatment due to continuously shifting paradigms to treat COVID-19 during the pandemic. The open label design has potential biases, however, the endpoints chosen, and outcomes collected by the researchers are highly objective outcomes, e.g., survival, need to transfer to ICU and days on ICU and we do not believe the design had a significant biasing effect on the outcome. Sudden high intake of iodine has been suggested to cause transient hypothyroidism in some instances, which is called the Wolff-Chaikoff effect, We would like to state that this effect is mainly described in persons with a far higher daily iodine intake than used in our trial and in patients with pre-existing thyroid conditions2 [22]. Since we excluded patients with thyroid diseases and the transient effect on TSH, T4 and T3 was not observed in multiple studies that included over 1000 individuals receiving a high dose of potassium iodine, we do not think this is a substantial bias or limitation in our study [22,23].

Next to the two primary endpoints mortality and ICU admission, we performed an exploratory analysis on the data gathered from the standardized WHO core CRFs. In this analysis, which was not corrected for multiple testing, we observed a significantly shorter stay at the ICU in the patients that were treated with both standard care and oral iodine. Of note, cautious interpretation is warranted because the study was not designed for investigating this endpoint and the number of patients included in this sub analysis is limited. These findings of course warrant replication. The pandemic passed, however, and left us in the absence of a replication cohort. Therefore, we set out to validate our findings *in vitro* by studying immunological responses in pulmonary cells in the presence of iodine. Here we observed that iodine empowers pulmonary tissue to exert a stronger IFN-I response, which indeed augmented the defense towards synthetic viruses.

Something to consider in our study is the timing of iodine administration. A recent finding from a study in lambs demonstrates that animals receiving iodine before infection with RSV exhibited the best pulmonary protection from infection [8]. It might very well be that adding iodine earlier in treatment, for instance at home when someone tests positive and only has mild symptoms, or ideally before infection, might exert a much stronger reduction in disease progression, resulting in less hospital admissions. This hypothesis is in line with our *in vitro* validation, where we observed enhanced IFN-I responses upon iodine treatment, which are crucial in the early phase of disease progression. However, by the time the patients were included in this study, these patients were severely diseased, most likely as a result from excessive systemic inflammation and cytokine production. It would therefore be of high interest to unravel what the effect of iodine in a highly inflammatory context is. Since IFN-I remains one of the most potent antiviral cytokines, we hypothesize that even in severe diseased patients that undergo excessive systemic inflammation, iodine-enhanced IFN-I responses remain beneficial. This hypothesis is supported by our clinical observation but warrants further investigation.

Besides the role of iodine in early, innate immune responses, literature suggests additional roles for iodine in secondary, adaptive immune responses. Accordingly, administering iodine might help to produce IgG in the secondary immune response, which is necessary to overcome the viral infection [9]. Besides, numerous studies show that severe worsening of COVID-19 symptoms occurs especially in patients who fail to mount an adequate secondary immune response, which hypothetically can be rescued with additional administration of iodine in patient groups with low basal levels. In addition, patients in which this secondary immune response is hampered, for instance by use of rituximab-induced B cell depletion treatment, have a significantly increased risk of severe illness and death by COVID-19 [24,25].

We believe, however, that our findings are worthwhile to further investigate, both clinical and mechanistic. For instance, it would be interesting to investigate whether patients receiving additional iodine can produce sufficient IgG levels to reduce viral loads more effectively and at earlier time points compared to patients without iodine supplementation. Speculatively, this could result in a shorter time of intensive care admission, in line with what we observed in this study. If this is the case, oral iodine can be a cost-effective treatment to reduce the burden of COVID-19 on patients and perhaps could be an agent of interest in future pandemics to come. Therefore, we advocate further research to elucidate the potential benefits of iodine treatment in viral infections.

## Supporting information

**S1 Fig. Representative microscopy images of smRNA FISH data and analysis on IFNβ, IFIT1, MX1 expression upon Lugol treatment and PolyIC transfection. A** IFNβ expression in Calu-3 lung epithelium. **B** automated image analyses of IFNβ spot detection and assignment to nearest nucleus. **C** IFIT1 expression in Calu-3 lung epithelium. **D** MX1 expression in Calu-3 lung epithelium.
(PNG)

**S2 Fig. Lugol pretreatment does not affect PolyIC transfection efficiencies. A** Representative microscopy image of Calu-3 lung epithelium transfected with rhodamine-labeled PolyIC using lipofectamine for 6 hours. **B** Representative microscopy image as in A. Cells were treated with 1% Lugol solution for 7 hours in total, starting 1 hour prior transfection with rhodamine-labeled PolyIC for 6 hours. **C** Quantification of rhodamine signal of 7 representative regions per condition, across n = 2.
(PNG)

**S3 Fig. Representative microscopy images of smRNA FISH data on IFNβ, IFIT1, MX1 expression upon Lugol treatment. A** IFNβ expression, close, if not zero, in Calu-3 lung epithelium. **B** IFIT1 expression in Calu-3 lung epithelium. **C** MX1 expression in Calu-3 lung epithelium.
(PNG)

**S1 File. C1 protocol translated.**
(PDF)

**S2 File. CONSORT 2025 flow diagram Trakseletal.**
(DOCX)

**S3 File. C1-protocol jodium RCT CCMO.**
(PDF)

**S4 File. CONSORT checklist Trakseletal.**
(DOCX)

**S5 File. Final dataset for plosone.**
(CSV)

## Acknowledgments

We would like to thank the patients for participating in this study and the nursing and medical staff for their unwaning determination in carrying this clinical trial forward.

## Author contributions

**Conceptualization:** René Traksel, Arjen van Henten, Laura van Eyndhoven, Richard Verheesen.

**Data curation:** Jasper Broen.

**Formal analysis:** Jasper Broen, Laura van Eyndhoven.

**Funding acquisition:** René Traksel.

**Investigation:** René Traksel, Arjen van Henten, Marc Königs, Laura van Eyndhoven.

**Methodology:** René Traksel, Marc Königs, Arjun Raj, Laura van Eyndhoven, Richard Verheesen.

**Resources:** Arjun Raj.

**Supervision:** Jasper Broen.

**Writing – original draft:** René Traksel, Jasper Broen, Laura van Eyndhoven.

**Writing – review & editing:** René Traksel, Jasper Broen, Laura van Eyndhoven, Richard Verheesen.

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
