## [Decision Letter · Decision Letter 0]

8 Oct 2025

Dear Dr. Broen,

We look forward to receiving your revised manuscript.

Kind regards,

Cheorl-Ho Kim, Ph.D.

Academic Editor

PLOS ONE

Journal Requirements:

“Maxima Medical Center Innovation Fund”

3. Please update your submission to use the PLOS LaTeX template. The template and more information on our requirements for LaTeX submissions can be found at http://journals.plos.org/plosone/s/latex .

“This study has been performed with financial aid of the Maxima Medical Center Innovation Fund for which we are grateful. “

“Maxima Medical Center Innovation Fund”

5. Please amend the manuscript submission data (via Edit Submission) to include authors R.A.M. Traksel MD, J.C.A. Broen MD, PhD , M.J. Henten MD , M.H.H Königs MD, PhD , A. Raj PhD , L.C. Van Eyndhoven PhD,*, R.H. Verheesen.

6. We notice that your supplementary figures are included in the manuscript file. Please remove them and upload them with the file type 'Supporting Information'. Please ensure that each Supporting Information file has a legend listed in the manuscript after the references list.

Additional Editor Comments (if provided):

Thank you for your submission to Plos One.

I have completed to review your manuscript, as you can read them.

I am very pleased to inform that your manuscript will be reconsidered for publication in Plos One.

One major revision comments are slightly critical but I think the criticisms can strengthen your manuscript if you would properly revise it.

I would like to receive your revision within one month.

Thank you

Sincerely

Cheorl-Ho Kim

Editor

Reviewers' comments:

Reviewer's Responses to Questions

**Comments to the Author**

1. Is the manuscript technically sound, and do the data support the conclusions?

Reviewer #1: Partly

Reviewer #2: Yes

2. Has the statistical analysis been performed appropriately and rigorously?

Reviewer #1: Yes

Reviewer #2: Yes

3. Have the authors made all data underlying the findings in their manuscript fully available?

Reviewer #1: No

Reviewer #2: Yes

4. Is the manuscript presented in an intelligible fashion and written in standard English?

Reviewer #1: Yes

Reviewer #2: Yes

Reviewer #1: Major Revisions

A. Thyroid Function & The Wolff-Chaikoff Effect: As a high-dose iodine intervention carries the risk of inducing transient hypothyroidism through the Wolff-Chaikoff effect, it would significantly improve the manuscript to include thyroid function test data. At a minimum, please discuss this known side effect and address the omission of thyroid monitoring as a limitation.

B. Statistical Power: The study's sample size of 141 is smaller than the target of 200, making the study underpowered for its primary outcomes. The authors should explicitly acknowledge that the failure to find a significant effect on mortality or ICU transfer may be due to this limitation.

C. Exploratory vs. Primary Outcomes: The manuscript highlights the positive finding for reduced ICU stay in the abstract and conclusion, but this was a secondary outcome. It is crucial to rephrase this to emphasize that the primary endpoints were not met, and the positive finding is exploratory and hypothesis-generating.

D. Open-Label Design: The unblinded, open-label design introduces a risk of bias, particularly for any subjective or patient-reported outcomes. Acknowledging this limitation in the discussion will provide a more transparent and robust account of the study.

Minor Revisions

e. Please replace the term "lege artis" with a specific description of the standard of care provided to the control group.

f. Clarify the patient enrollment process, specifically how patients who were too ill to consent were handled, to ensure transparency regarding potential selection bias.

g. The CONSORT flowchart indicates that some patients were excluded from the analysis. This appears to conflict with the statement that "We did not encounter missing data." Please reconcile this discrepancy.

h. Please elaborate within Methods how Renal replacement therapy was calculated (0.00 vs. 0.01 in the intervention group as stated in Table 2).

i. The statistical methods section would be clearer with a more detailed breakdown of which tests were applied to which specific variables.

j. Confirm that all figures and tables are consistently formatted, with clear, descriptive legends.

k. Ensure consistent use of terminology for the iodine intervention, either "potassium iodine tablets" or "Lugol's solution" throughout the manuscript.

Reviewer #2: Iodine does not reduce mortality or ICU admissions; it may enhance antiviral immunity through increased type I interferon responses, contributing to shorter ICU stays in COVID-19 patients—the role of iodine in enhancing IFN-I mediated antiviral immunity. This article is of interest. The authors have new results, and they are well written. Therefore, this article is suitable for publication.

**Do you want your identity to be public for this peer review?** For information about this choice, including consent withdrawal, please see our Privacy Policy

Reviewer #1: No

Reviewer #2: No

---

## [Author Response · Author response to Decision Letter 1]

12 Dec 2025

Eindhoven, The Netherlands 8th of November

Concerns: Revised version of manuscript PONE-D-25-32733

Dear editor, Dear Dr. Cheorl-Ho Kim,

Thank you for considering the manuscript “Iodine increases type I interferon responses and decreases covid-19 disease severity: results from an open label randomized clinical trial” by Traksel et al. for publication in PLOS ONE after careful revision.

Hereby we would like to provide the journal with the rebuttal letter in response to the issues raised by the reviewers. In addition to the revised manuscript, we added a revised manuscript with tracked changes as requested.

We thank the editor for pointing towards a possibility to make all laboratory protocols available for the public. We are glad to inform the editor that all used laboratory and imaging analyses protocols have been published and are listed in the references.

With kind regards,

On behalf of all authors,

Jasper Broen MD, PhD

Rheumatologist and Clinical Pharmacologist

Response to Reviewers

Response to academic editor

The academic editor considers the manuscript for publication after proper revision and mentions that the criticisms of the reviewers provide a possibility to further strengthen the manuscript.

We would like to thank the editor for these kind words and hereby provide our thoughtful response to the raised issues and comments.

1. The academic editor requests to format the manuscript files in accordance with PLOS ONE’s style requirements.

We have now adjusted the manuscripts files according to the provided PLOS ONE templates.

2. The academic editor asks to state the role of the “Maxima Medical Center Innovation Fund” and add a Role of Funder statement in the cover letter.

We thank the academic editor for pointing this out and indeed stated: “The funders had no role in study design, data collection and analysis, decision to publish, or preparation of the manuscript.” We appreciate the effort provided by the editorial office to change the submission form for us.

3. The academic editor requests to update the submission by using the PLOS LaTeX template.

We updated the submission as requested.

4. The editor mentions a discrepancy between the statements in the Acknowledgements section and the Funding Statement.

We have now updated the submission as requested and would like to thank the editorial office for changing the online submission form on our behalf.

5. The editor requests to amend the manuscript submission data to include all authors stated on the manuscript.

We have completed the manuscript submission data accordingly.

6. It is noticed that the supplementary figures are included in the manuscript file, these are directed to upload separately in a file type “supporting information”.

We have now removed the supplementary files from the manuscript and uploaded the figures separately.

7. The editor advises to evaluate any specific citations suggested by the reviewers for relevancy before adding them to the manuscript.

We thank the editor for this advice. We evaluated the suggestion made by reviewer 2 carefully.

8. The editor advises to review the references carefully and to mention changes in the tracked changes file.

We have reviewed the references carefully and highlighted the changes in the manuscript.

Response to peer reviewers comments.

1. Is the manuscript technically sound and do the data support the conclusions?

Reviewer 1# Partly

Reviewer 2# Yes

We thank the reviewers for their time and comments, reviewer 1 further eludes on this verdict in 5.B.C.D. within the paragraph “Review Comments to the Author”. We added a more elaborate answer to these issues accordingly in this paragraph below. We would like to thank reviewer 2 for the positive answer.

2. Has the statistical analysis been performed appropriately and rigorously?

Reviewer 1# Yes

Reviewer 2# Yes

We thank the reviewers for their positive feedback.

3. Have the authors made all data underlying the findings in their manuscript fully available?

Reviewer #1: No

Reviewer #2: Yes

We thank reviewer #2 for the positive feedback and provided an explanation on how to get access to the full data. We like to point out that the requested data points behind means, medians and variance measures for the positive findings are already directly appreciable from the manuscript figures.

4. Is the manuscript presented in an intelligible fashion and written in standard English?

Reviewer 1# Yes

Reviewer 2# Yes

We thank both reviewers for their positive feedback

5. Review comments to the author.

Review comments to the author

Reviewer #1: Major Revisions

A. The reviewer advises to include thyroid function test data since a high-dose iodine intervention carries the risk of inducing transient hypothyroidism by the Wolff-Chaikoff effect. Or if not available, asks us to at least discuss this effect and address this as a limitation.

We thank the invitation of the reviewer to further elude on the Wolff-Chaikoff effect. We would like to state that this effect is mainly described in persons with a far higher daily iodine intake than used in our trial and in patients with pre-existing thyroid conditions1. Since we excluded patients with thyroid diseases and the transient effect on TSH, T4 and T3 was not observed in multiple studies that included over 1000 individuals receiving a high dose of potassium iodine, we do not think this is a substantial bias or limitation in our study1,2. TSH and FT4 levels were not routinely checked after administration of the potassium iodine as part of the study protocol, however in clinical care setting at the ICU these values were tested in 30% of the patients and no signs of hypothyroidism were present. As suggested by the reviewer, we have now added this to the discussion.

1. Markou K, Georgopoulos N, Kyriazopoulou V, Vagenakis AG. Iodine-Induced hypothyroidism. Thyroid. 2001 May;11(5):501-10. doi: 10.1089/105072501300176462. PMID: 11396709.

2. Nauman J, Wolff J. Iodide prophylaxis in Poland after the Chernobyl reactor accident: benefits and risks. Am J Med. 1993 May;94(5):524-532. doi: 10.1016/0002-9343(93)90089-8. PMID: 8498398.

B. The reviewer mentions that the sample size is smaller than the target size and the failure to find a significant effect on mortality or ICU transfer might be due to this limitation. The reviewer requests to explicitly acknowledge this in the manuscript.

We agree with the reviewer and added to the discussion:

“Notably, due to the pandemic coming to an end we were not able to reach the target size, which might have played a role in failure to find a significant effect on mortality or ICU transfer.”

C. The reviewer asks to emphasize that the primary endpoints were not met and the secondary finding is exploratory and hypothesis-generating.

The discussion included the following sentences:

“Our primary hypothesis was that oral iodine reduces mortality and reduces transfers to the intensive care unit. Our study does not corroborate this hypothesis.” We changed this to “Our study does not corroborate this hypothesis and primary endpoints were not met.”

Furthermore the discussion already included:

“Next to the two primary endpoints mortality and ICU admission, we performed an exploratory analysis on the data gathered..”

“Of note, cautious interpretation is warranted because the study was not designed for investigating this endpoint and the number of patients included in this sub analysis is limited. These findings of course warrant replication.”

We hope this addresses the concerns of the reviewer sufficiently.

D: The reviewer asks us to acknowledge the limitations of an unblinded, open label design in the discussion.

This limitation has been addressed in the current discussion, it reads “The open label design has potential biases, however, the endpoints chosen, and outcomes collected by the researchers are highly objective outcomes e.g. survival, need to transfer to ICU and days on ICU and we do not believe the design had a significant biasing effect on the outcome.”

Minor revisions

E: The reviewer asks to replace the term lege artis with a specific description of the standard of care provided to the control group.

We followed the advice of the reviewer and have now replaced the term with a more specific description. We would like to add that the issue of changing treatment throughout the pandemic is tackled with the block randomization procedure.

F. The reviewer asks us to clarify how patients that were too ill to consent were handled to ensure transparency regarding potential selection bias.

All patients were included at the ward and were able to provide consent themselves, this is pointed out in the “Methods” section.

G. The reviewer mentions that some patients were excluded from the analysis based on the flow-chart and mentions that this is in conflict with the statement “we did not encounter missing data”.

We do not believe this is in conflict, patients that were lost to follow-up or did not fulfil the full dosing scheme were excluded from analysis as is appreciable from the flow-chart. There were no missing data in the groups ultimately analysed for the primary outcomes. In the results section the reason for excluding patients in the analyses is stated.

H. The reviewer asks us to elaborate within the Methods how Renal replacement therapy was calculated.

Table 2 states the proportion of patients that had a certain outcome. Of the 67 patients that received iodine treatment, one patient received renal replacement therapy. This fraction is 0.0149 which was rounded to two decimals in table formatting and leads to a proportion of 0.01. Of course the statistics have been performed on the unrounded numbers. If we provide this information in the methods section for this variable, we need to do this for all variables and we do not believe this level of detail adds to the readability level of the Methods section. To provide more clarity on the matter, we added in the legend that numbers are rounded in two decimals behind the comma.

I. The reviewer asks to provide a breakdown of which tests were applied to which specific variables.

For continuous variables (e.g. days of admission, age) we used t-tests and linear regression and for non-continuous variables such as sex, mortality we used chi-square tests or Fisher exact tests where appropriate. We added this to the Methods section.

J. The reviewer asks to confirm that all figures and tables are consistently formatted with clear, descriptive legends.

All figure and tables were formatted following the Latex template.

K. The reviewer asks to be consistent in the use of terminology for the iodine intervention and either use potassium iodine or Lugols solution throughout the manuscript.

We have clarified the definition of Lugol solution by adding the following information to the revised manuscript: “To investigate whether iodine modulates antiviral immunity, we pretreated cells with Lugol’s solution, a ready-to-use aqueous solution containing 0.7% potassium iodide and 0.35% iodine dissolved in water (Sigma-Aldrich, 32922). We treated the cells at a concentration range of 0.01%-1% (v/v) Lugol’s solution in culture medium, which corresponds to a iodine concentration of 0.35-35µg/mL.”

Reviewer #2

1. The reviewer mentions that our article is of interest, contains new results and is well written and provides the advice to publish it.

We would like to thank the reviewer for this positive feedback and the recommendation for publication.

---

## [Decision Letter · Decision Letter 1]

4 Jan 2026

Iodine increases pulmonary type I interferon responses and decreases covid-19 disease severity: results from an open label randomized clinical trial.

PONE-D-25-32733R1

Dear Dr. Brien,

We’re pleased to inform you that your manuscript has been judged scientifically suitable for publication and will be formally accepted for publication once it meets all outstanding technical requirements.

Kind regards,

Cheorl-Ho Kim, Ph.D.

Academic Editor

PLOS One

Additional Editor Comments (optional):

Dear Dr Broen,

I would like to express my deep appreciation for your submission and revision as well as for your patience in waiting for our review and editorial process.

I am confident that the careful dealing with manuscripts is the most important point to document the scientific findings, even not in leading journals like nature and science as a medium.

That is the reason what I have had a long time to decide the final direction. Your present revision is now acceptable in the present form.

Congratulations on your study.

Thank you again

Sincerely

Cheorl-Ho Kim PhD

Editor

Professor

Dept Biological Sciences

Sungkyunkwan University

Korea

Reviewers' comments:

Reviewer's Responses to Questions

**Comments to the Author**

Reviewer #1: All comments have been addressed

2. Is the manuscript technically sound, and do the data support the conclusions?

Reviewer #1: Yes

3. Has the statistical analysis been performed appropriately and rigorously?

Reviewer #1: N/A

4. Have the authors made all data underlying the findings in their manuscript fully available?

Reviewer #1: Yes

5. Is the manuscript presented in an intelligible fashion and written in standard English?

Reviewer #1: Yes

Reviewer #1: None. The authors address all of my comments (including the Major and Minor comments) and at this point I've got no further comments.

**Do you want your identity to be public for this peer review?** For information about this choice, including consent withdrawal, please see our Privacy Policy

Reviewer #1: **Yes:** Dr Yaniv. S. Ovadia RD PhD

---

## [Editor Report · Acceptance letter]

PONE-D-25-32733R1

PLOS One

Dear Dr. Broen,

I'm pleased to inform you that your manuscript has been deemed suitable for publication in PLOS One. Congratulations! Your manuscript is now being handed over to our production team.

Kind regards,

on behalf of

Professor Cheorl-Ho Kim

Academic Editor

PLOS One